# Why do deep convolutional networks generalize so poorly to small image transformations?

## Abstract

Deep convolutional network architectures are often assumed to guarantee generalization for small image translations and deformations. In this paper we show that modern CNNs (VGG16, ResNet50, and InceptionResNetV2) can drastically change their output when an image is translated in the image plane by a few pixels, and that this failure of generalization also happens with other realistic small image transformations. Furthermore, we see these failures to generalize more frequently in more modern networks. We show that these failures are related to the fact that the architecture of modern CNNs ignores the classical sampling theorem so that generalization is not guaranteed. We also show that biases in the statistics of commonly used image datasets makes it unlikely that CNNs will learn to be invariant to these transformations. Taken together our results suggest that the performance of CNNs in object recognition falls far short of the generalization capabilities of humans.

## 1 Introduction

Deep convolutional neural networks (CNNs) have revolutionized computer vision. Perhaps the most dramatic success is in the area of object recognition, where performance is now described as "superhuman" (He et al., 2015). A key to the success of any machine learning method is the *inductive bias* of the method, and clearly the choice of architecture in a neural network significantly affects the inductive bias. In particular, the choice of convolution and pooling in CNNs is motivated by the desire to endow the networks with invariance to irrelevant cues such as image translations, scalings, and other small deformations (Fukushima & Miyake, 1982; Zeiler & Fergus, 2014). This motivation was made explicit in the 1980s by Fukushima in describing the "neocognitron" architecture, which served as inspiration for modern CNNs (LeCun et al., 1989), "After finishing the process of learning, pattern recognition is performed on the basis of similarity in shape between patterns, and is not affected by deformation, nor by changes in size, nor by shifts in the position of the input patterns." (Fukushima, 1988)

Despite the excellent performance of CNNs on object recognition, the vulnerability to adversarial attacks suggests that superficial changes can result in highly non-human shifts in prediction (e.g. (Bhagoji et al., 2017; Su et al., 2017). In addition, filtering the image in the Fourier domain (in a way that does not change human prediction) also results in a substantial drop in prediction accuracy (Jo & Bengio, 2017). These and other results (Rodner et al., 2016) indicate that CNNs are not invariant to cues that are irrelevant to the object identity.

An argument against adversarial attacks on CNNs is that they often involve highly unnatural transformations to the input images, hence in some sense we would not expect CNNs to be invariant to these transformations. When considering more natural transformations, there is preliminary evidence that AlexNet (Krizhevsky et al., 2012) is robust to some of them (Zeiler & Fergus, 2014). On the other hand, there is also preliminary evidence for lack of robustness in the more modern networks for object classification (Bunne et al., 2018) and detection (Rosenfeld et al., 2018) along with studies suggesting that with small CNNs and the MNIST data, data augmentation is the main feature affecting CNN invariance (Kauderer-Abrams, 2017). An indirect method to probe for invariances measures the linearity of the learned representations under natural transformations to the input image (Lenc

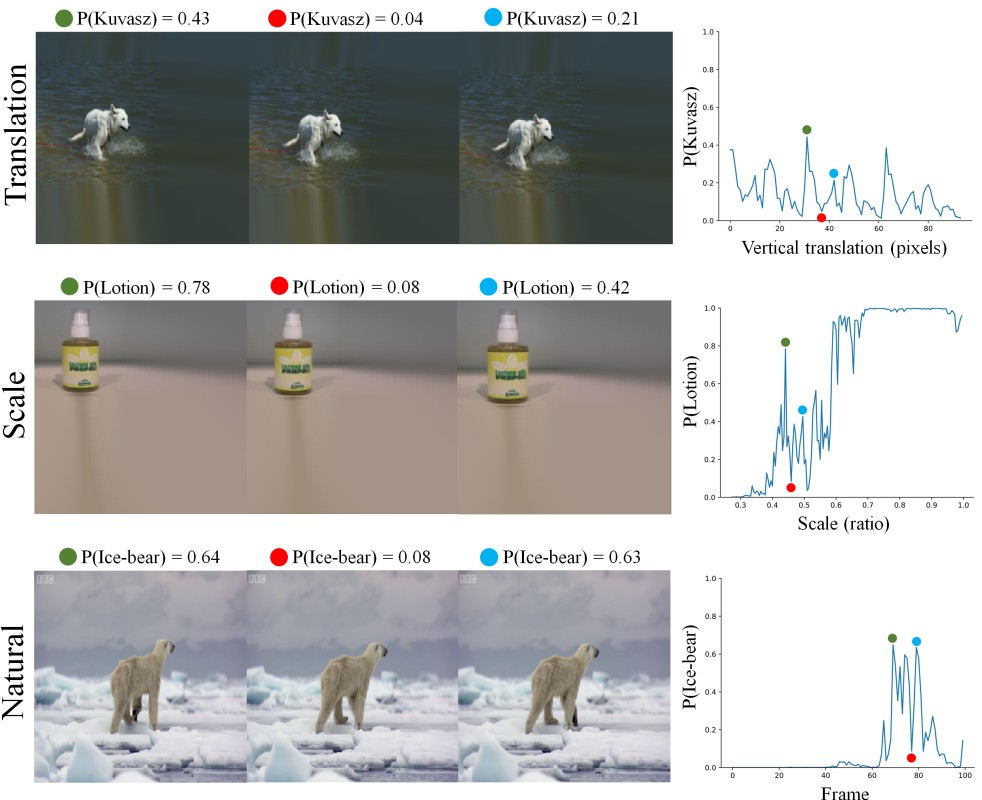

Figure 1: Examples of jagged predictions of modern deep convolutional neural networks. Top: A negligible vertical shift of the object (Kuvasz) results in an abrupt decrease in the network's predicted score of the correct class. Middle: A tiny increase in the size of the object (Lotion) produces a dramatic decrease in the network's predicted score of the correct class. Bottom: A very small change in the bear's posture results in an abrupt decrease in the network's predicted score of the correct class. Colored dots represent images chosen from interesting x-axis locations of the graphs on the right. These dots illustrate sensitivity of modern neural networks to small, insignificant (to a human), and realistic variations in the image.

& Vedaldi, 2015; Hénaff & Simoncelli, 2015; Fawzi & Frossard, 2015; Cohen & Welling, 2014). The recent work of (Engstrom et al., 2017) investigates adversarial attacks that use only rotations and translations. They find that "simple transformations, namely translations and rotations alone, are sufficient to fool neural network-based vision models on a significant fraction of inputs" and show that advanced data augmentation methods can make the networks more robust.

In this paper, we directly ask "why are modern CNNs not invariant to natural image transformations despite the architecture being explicitly designed to provide such invariances?". Specifically, we systematically examine the invariances of three modern deep CNNs: VGG-16 (Simonyan & Zisserman, 2014), ResNet-50 (He et al., 2016), and InceptionResNet-V2 (Szegedy et al., 2017). We find that modern deep CNNs are not invariant to translations, scalings and other realistic image transformations, and this lack of invariance is related to the subsampling operation and the biases contained in image datasets.

## 2 FAILURES OF MODERN CNNS

Figure 1 contains examples of abrupt failures following tiny realistic transformations for the InceptionResNet-V2 CNN. Shifting or scaling the object by just one pixel could result in a sharp change in prediction. In the top row, we embed the original image in a larger image and shift it in the

image plane (while filling in the rest of the image with a simple inpainting procedure). In the middle row, we repeat this protocol with rescaling. In the bottom row, we show frames from a BBC film in which the ice bear moves almost imperceptibly between frames and the network's output changes dramatically[1].

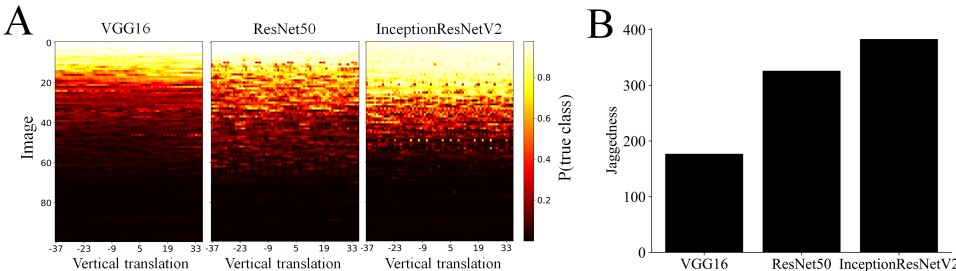

Figure 2: Modern deep convolutional neural networks are sensitive to small image translations. A) Comparison of three networks of various depths in the task of vertical image translation depicted in figure 1. Images (rows) are randomly chosen from the ImageNet dataset (Deng et al., 2009), and are sorted by the network's prediction sum in a descending order. B) More modern networks have more jagged predictions.

In order to measure how typical these failures are, we randomly chose images from the ImageNet validation set and measured the output of three modern CNNs as we embedded these images in a larger image and systematically varied the vertical translation. As was the case in figure 1, we used a simple inpainting procedure to fill in the rest of the image.

Results are shown in figure 2. Each row corresponds to an image under different translations and the color denotes the network's estimate of the probability of the correct class. Thus a row that is all light corresponds to a correct classification that is invariant to translation, while a row that is all dark corresponds to an incorrect classification that is invariant to translation. Surprisingly, many rows show abrupt transitions from light to dark, indicating that the classification changes abruptly as the object is translated. We quantify the lack of invariance by a measure we call "jaggedness": the number of times the network's predictions had the correct class in its top-5 and after just one pixel shift it moved outside of the top-5 (and also the opposite transition from non-top-5 to top5). Using this measure, we find that for approximately 30% of the images, the output is "jagged", i.e the network changes its prediction by a shift of a single pixel. Also, as shown in the right of figure 2, jaggedness is greater for the modern, deeper, networks compared to the less modern VGG16 network. While the deeper networks have better test accuracy, they are also less invariant. In the appendix we also show an alternative to the "jaggedness" measure, which gives similar results.

A natural criticism of these results is that they are somehow related to the image resizing and inpainting procedures that we used. To test this possibility, we repeated the experiment with a different protocol where we chose different crops of the original ImageNet image while making sure that the object bounding box remained within the crop. This protocol does not require any inpainting while still translating the object location within the new image. Results are shown in the appendix. We still have a large fraction of images for which the prediction is not invariant to translation. We also show in the appendix similar results for scaling rather than translation. Overall we find that regardless of the protocol used, modern CNNs often change their output significantly as a result of a small translation or scaling.

## 3 IGNORING THE SAMPLING THEOREM

The failure of CNNs to generalize to image translations is particularly puzzling. Intuitively, it would seem that if all layers in a network are convolutional then the representation should simply translate when an image is translated. If the final features for classification are obtained by a global pooling

---

[1]Frames were extracted from the film and presented to the network using the pipeline described in the appendix. Different pipelines give different results.

operation on the representation (as is done for example in ResNet50 and InceptionResNetV2) then these features should be invariant to translation. Where does this intuition fail?

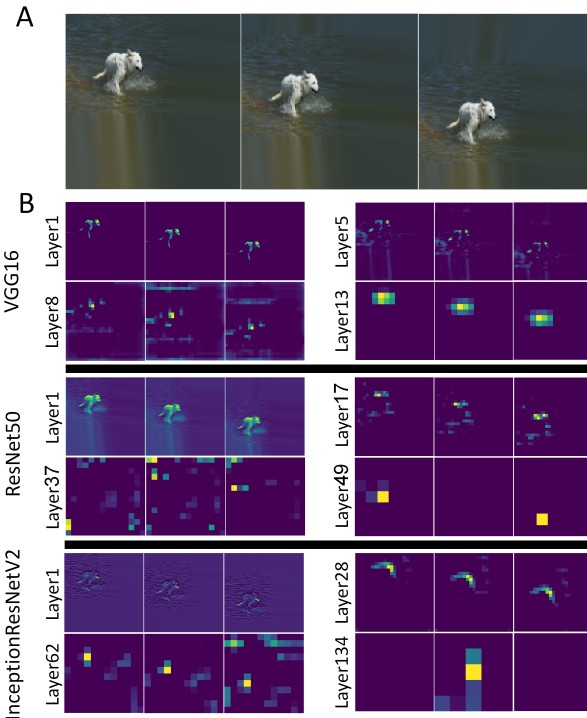

Figure 3: The deeper the network, the less shiftable are the feature maps. A) A vertical shift of a "Kuvasz" dog in the image plane. B) Feature maps from three different network architectures in response to the translated Kuvasz image. Layer depth assignments reflect the number of trainable convolutional layers preceding the selected layer. The last layer is always the last convolutional layer in each network.

This intuition ignores the subsampling operation which is prevalent in modern CNNs, also known as "stride". This failure of translation invariance in systems with subsampling was explicitly discussed in Simoncelli et al. (Simoncelli et al., 1992) who wrote "We cannot literally expect translation invariance in a system based on convolution and subsampling: translation of the input signal cannot produce simple translations of the transform coefficients, unless the translation is a multiple of each of the subsampling factors in the system". Since deep networks often contain many subsampling operations, the subsampling factor of the deep layers may be very large so that "literal" translation invariance only holds for very special translations. In InceptionResnetV2, for example, the subsampling factor is 60, so we expect exact translation invariance to hold only for $\frac{1}{60^2}$ of possible translations.

Simoncelli et al. also defined a weaker form of translation invariance, which they called "shiftability" and showed that it can hold for systems with subsampling (this is related to weak translation invariance as defined by (Lenc & Vedaldi, 2015), see also (Esteves et al., 2017; Cohen & Welling, 2014) for related ideas applied to neural networks). Here we extend the basic shiftability result to show that when shiftability holds, then global pooling will indeed yield invariant representations.

We define $r(x)$ as the response of a feature detector at location $x$ in the image plane. We say that this response is "convolutional" if translating the image by any translation $\delta$ yields a translation of the response by the same $\delta$. This definition includes cases when the feature response is obtained by convolving the input image with a fixed filter, but also includes combinations of linear operations and nonlinear operations that do not include any subsampling.

We start by a trivial observation:

**Observation:** If $r(x)$ is convolutional then global pooling $r = \sum_x r(x)$ is translation invariant.

**Proof:** This follows directly from the definition of a convolutional response. If $r(x)$ is the feature response to one image and $r_2(x)$ is the feature response to the same image translated, then $\sum_x r(x) = \sum_x r_2(x)$ since the two responses are shifts of each other.

**Definition:** A feature detector $r(x)$ with subsampling factor $s$ is called "shiftable" if for any $x$ the detector output at location $x$ can be linearly interpolated from the responses on the sampling grid:

$$r(x) = \sum_i B_s(x - x_i) r(x_i)$$

where $x_i$ are located on the sampling grid for subsampling factor $s$ and $B_s(x)$ is the basis function for reconstructing $r(x)$ from the samples.

The classic Shannon-Nyquist theorem tells us that $r(x)$ will be shiftable if and only if the sampling frequency is at least twice the highest frequency in $r(x)$.

**Claim:** If $r(x)$ is shiftable then global pooling on the sampling grid $r = \sum_i r(x_i)$ is translation invariant.

**Proof:** This follows from the fact that global pooling on the sampling grid is (up to a constant) the same as global pooling for all $x$.

$$
\begin{aligned}
\sum_x r(x) &= \sum_x \sum_i r(x_i) B(x - x_i) & (1) \\
&= \sum_i r(x_i) \sum_x B(x - x_i) & (2) \\
&= K \sum_i r(x_i) & (3)
\end{aligned}
$$

where $K = \sum_x B(x - x_i)$ and $K$ does not depend on $x_i$.

While the claim focuses on a global translation, it can also be extended to piecewise constant transformations.

**Corollary:** Consider a set of transformations $T$ that are constant on a set of given image subareas. If $r(x)$ is shiftable and for a given image, the support of $r(x)$ and its receptive field is contained in the same subregion for all transformations in $T$, then global pooling on the sampling grid is invariant to any transformation in $T$.

**Proof:** This follows from the fact that applying any transformation in $T$ to an image has the same effect on the feature map $r(x)$ as translating the image.

To illustrate the importance of the sampling theorem in guaranteeing invariance in CNNs, consider a convolutional layer in a deep CNN where each unit acts as a localized "part detector" (this has been reported to be the case for many modern CNNs (Zeiler & Fergus, 2014; Zhou et al., 2014)). Each such part detector has a spatial tuning function and the degree of sharpness of this tuning function will determine whether the feature map can be subsampled while preserving shiftability or not. For example, consider a part detector that fires only when the part is exactly at the center of its receptive field. If there is no subsampling, then as we translate the input image, the feature map will translate as well, and the global sum of the feature map is invariant to translation. But if we subsample by two (or equivalently use a stride of two), then there will only be activity in the feature map when the feature is centered on an even pixel, but not when it is centered on an odd pixel. This means that the global sum of the feature map will *not* be invariant to translation.

In the language of Fourier transforms, the problem with a part detector that fires only when the part is exactly at the center of the receptive field is that the feature map contains many high frequencies and hence it cannot be subsampled while preserving shiftability. On the other hand, if we have a part detector whose spatial tuning function is more broad, it can be shiftable and our claim (above) shows that the global sum of activities in a feature map will be preserved for all translations, even though the individual firing rates of units will still be different when the part is centered at an odd pixel or an even pixel. Our corollary (above), shows the importance of shiftability to other smooth transformations: in this case each "part detector" will translate with a *different* translation but it is still the case that nonshiftable representations will not preserve the global sum of activities as the image is transformed, while shiftable representations will.

Figure 3 examines the extent to which the representations learned by modern CNNs are invariant or shiftable. The top row shows an image that is translated vertically, while the bottom three rows show the representations in different layers for the three CNNs we consider. For VGG16 the representation appears to shift along with the object, including the final layer where the blurred pattern of response is not a simple translation of the original response, but seems to preserve the global sum for this particular image. For the two more modern networks, the responses are sharper but lose their shiftability in the later layers. In particular, the final layers show approximate invariance to one special translation but no response at all to another translation, suggesting that the many layers of subsampling yield a final response that is not shiftable.

We also performed a more quantitative measure of shiftability by counting for a given image the number of times the global sum of activities in each layer changes significantly (more than 20% of mean) as the input is shifted (for each image, we only considered feature maps where the maximum response was above a threshold). We call this measure "nonshiftability". According to the preceding analysis, in architectures that obey the sampling theorem, the global sum should be invariant to input translation so nonshiftability should be zero in all layers. We find that for all three networks, the initial layers have nonshiftability close to zero but as we go deeper and deeper nonshiftability increases. Furthermore, the deeper, more modern networks, exhibit larger nonshiftability in their deep layers compared to VGG16 (see appendix for graphs).

How can we guarantee that representations in CNNs will be shiftable? As explained above, we need to make sure that any feature map that uses stride does not contain frequencies above the Nyquist frequency. If CNNs were purely linear, we could simply blur the input images so that they would not include any frequencies higher than the Nyquist limit determined by the final sampling factor of the network. But since CNNs also include nonlinearities, they can add high frequencies that were not present in the input.

An important message of the sampling theorem is that *you should always blur before subsampling*. Translated to the language of neural networks this means that stride (i.e. subsampling) should always be combined with pooling (i.e. blurring) in the preceding layer. Indeed if we have an arbitrarily deep CNN where all the layers use stride=1 followed by one layer that has a stride greater than one, then by choosing the pooling window appropriately we can guarantee that the final layer will still be shiftable. If we do not use appropriate pooling then there is no guarantee that this layer will be shiftable. Even if we use appropriate pooling that ensures that a given layer is shiftable, the subsequent nonlinearities in a CNN may not preserve the shiftability, as the nonlinearities may again introduce high frequencies.

To illustrate the effect of pooling on shiftability in modern CNNs we replaced the $2 \times 2$ max pooling layers of VGG16 with $6 \times 6$ average pooling. This has the effect of reducing low freqencies but given the nonlinearities, it does not guarantee shiftability. As shown in figure 4 this simple change makes the representations approximately shiftable and, as predicted by our theory, the global sum is now invariant to both translations and rescalings of the input. This invariance of course comes with a price: the feature maps now have less detail and in preliminary experiments we find that recognition performance decreases somewhat. But the sampling theorem tells us that if we want to use subsampling while avoiding aliasing, we need to ensure that no high frequencies (relative to the Nyquist frequency) are present in the feature maps.

As an alternative to pooling, Ruderman et al. (Ruderman et al., 2018) have shown that networks may learn smooth filters that will lead to reduced sensitivity to transformations. Evidently, the filters learned in standard VGG16 are not smooth enough.

## 4    Why don't modern CNNs learn to be invariant from data?

While the preceding discussion suggests that the CNN architecture will not yield translation invariance "for free", there is still the possibility that the CNN will learn a translation invariant prediction from the training examples. This requires that the training set will actually be invariant to the irrelevant transformations. We examined the degree of invariance in the ImageNet training set by manually labeling the training images in five categories: Tibetan terrier, elephant, pineapple, bagel and rowing paddle.

Consistent with previous results on "dataset bias" (Simon et al., 2007; Raguram & Lazebnik, 2008; Berg & Berg, 2009; Torralba & Efros, 2011; Weyand & Leibe, 2011; Mezuman & Weiss, 2012) we

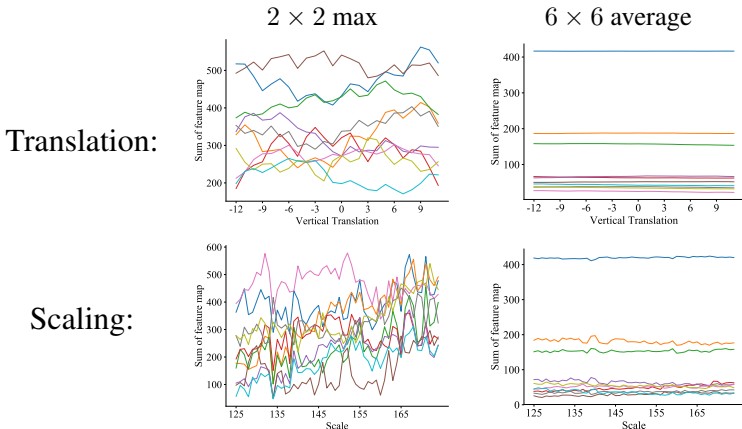

Figure 4: Average pooling makes VGG representations approximately shiftable. Plotted are summation of feature maps from the last layer of VGG (spatial dimension 7x7) as an input image is vertically translated in the image plane (top), or rescaled (bottom). Left: the original VGG16 with its 2x2 max pooling layers. Right: VGG16 where every 2x2 max pooling layer was replaced by a 6x6 average pooling layer. More randomly selected images are shown in the appendix.

find that the ImageNet dataset is not invariant to translations and rescalings. Figure 5 shows the distribution of the distances between the eyes of a "Tibetan terrier" and the positions of the center point between the dog's eyes. Notice that both distributions are far from uniform. Similar results are obtained for the other four categories.

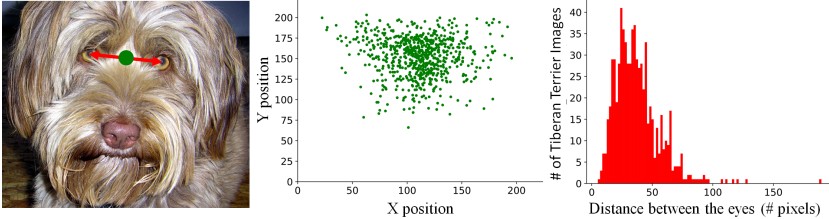

Figure 5: Photographer's biases in the ImageNet's "Tibetan terrier" category. Left: Example of the hand-labeling procedure. Middle: Positions of the middle point between the dog's eyes. Right: Histogram of distances between the dog's eyes. Notice the bias in both the object's position and scale.

To be more quantitative, we used the available bounding-box labels, and extracted the center point of the bounding-box and its height as proxies for the object position and size respectively. We then applied a statistical significance test to ask whether object location and object sizes were uniform for that category. For more than 900 out of the 1000 categories we found that location and size were highly non uniform ($P < 10^{-10}$). Given these strong biases, we cannot expect a learning system to learn to be invariant.

Even if the training set is not invariant, we can make it invariant using *data augmentation*. Will this make the CNN learn an invariant prediction? First, we note that we used pretrained networks and according to the authors' description of the training procedure, all three networks were trained using data augmentation. Obviously, not any data augmentation is sufficient for the networks to learn invariances. To understand the failure of data augmentation, it is again instructive to consider the subsampling factor. Since in modern networks the subsampling factor is approximately 60, then for a system to learn complete invariance to translation only, it would need to see $60^2 = 3600$ augmented versions of each training example, or it would need to have an inductive bias that allows it to generalize over transformations. If we also add invariance to rotations and scalings, the number grows exponentially with the number of irrelevant transformations. Engstrom et al. (Engstrom et al., 2017) suggest a sophisticated data augmentation method and show that it increases the invariance to

translation and rotation. However, for challenging datasets such as ImageNet the lack of invariance largely persists.

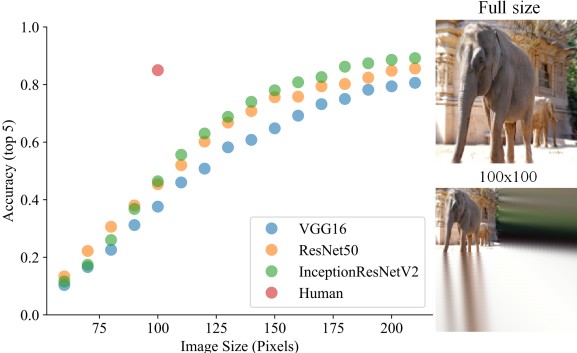

Figure 6: The performance of modern CNNs on test images from ImageNet that are embedded in a random location in a larger image is quite poor (less than 50% accuracy). Human performance is not affected. Right: An example of a full sized image and the same image resized to 100x100.

## 5    Implications for practical systems

Although our results show that modern CNNs fail to generalize for small image transformations, their performance on the ImageNet test set is still amazingly good and far better than previous techniques. This is related to the fact that the ImageNet test set contains the same photographer's biases as the training set, so generalization to very different sizes and locations is not required. To highlight this point, we created a new test set in which ImageNet images were embedded in a larger image in a random location (and the missing pixels were filled in using a simple inpainting algorithm). Figure 6 shows that human performance is not affected by the rescaling and random translations, while the performance of modern CNNs deteriorates dramatically. In fact, when images are scaled to half their original size and randomly translated, the accuracy of modern CNNs is less than 50%, typically considered poor performance.

One way in which modern systems partially address the lack of invariance is using *test time augmentation* in which the system output on a given image is computed by a majority vote among many random crops of the image. Clearly this is wasteful in resources and still only provides partial invariance.

## 6    Discussion

CNN architectures were designed based on an intuition that the convolutional structure and pooling operations will give invariance to translations and small image deformations "for free". In this paper we have shown that this intuition breaks down once subsampling, or "stride" is used and we have presented empirical evidence that modern CNNs do not display the desired invariances since the architecture ignores the classic sampling theorem. This still leaves open the possibility of a CNN learning invariance from the data but we have shown that the ImageNet training and testing examples include significant photographer's bias so that it is unlikely that a system will learn invariance using these examples.

In addition to pointing out these failures, the sampling theorem also suggests a way to impose translation invariance by ensuring that all representations are sufficiently blurred to overcome the subsampling. However, such blurred representations may lead to a decrease in performance, especially in datasets and benchmarks that contain photographer's bias. Alternatively, one could use specially designed features in which invariance is hard coded or neural network architectures that explicitly enforce invariance (Sifre & Mallat, 2013; Gens & Domingos, 2014; Cheng et al., 2016a;b; Dieleman et al., 2016; 2015; Xu et al., 2014; Worrall et al., 2017; Cohen & Welling, 2016). Again, as long as the datasets contain significant photographer's bias, such invariant approaches may lead to a decrease in performance.

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

APPENDIX

## A    PIPELINE FOR PRODUCING THE BOTTOM ROW OF FIGURE 1

We download this video: https://www.youtube.com/watch?v=0mgnf6t9VEc using an online downloader. We load the video frames and resize them to 299 by 299 as used by the standard Keras applications framework (https://keras.io/applications/). We preprocess the frames using the standard Keras preprocessing function. Finally, we use the predictions of the InceptionV3 model to demonstrate the jagged behavior shown in figure 1.

## B    OTHER SUPPLEMENTARY MATERIAL

| Network | Top-1 | Top-5 | Parameters | Depth |
|---|---|---|---|---|
| VGG16 (Simonyan & Zisserman, 2014) | 0.715 | 0.901 | 138,357,544 | 16 |
| ResNet50 (He et al., 2016) | 0.759 | 0.929 | 25,636,712 | 50 |
| InceptionResNetV2 (Szegedy et al., 2017) | 0.804 | 0.953 | 55,873,736 | 134 |

Table 1: The networks used (taken from (`https://keras.io/applications/`))

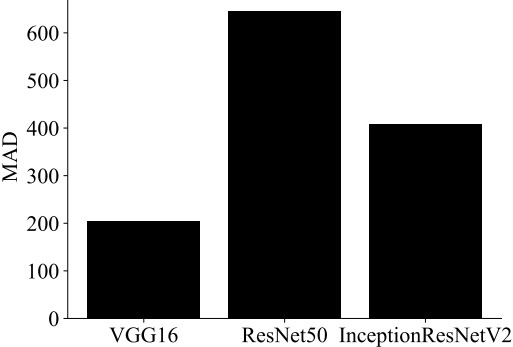

Figure 7: We also measure jaggedness using the Mean Absolute Difference (MAD) in the probability of the correct response as the image is shifted by a single pixel. Results are similar to those using the jaggedness measure described in the text.

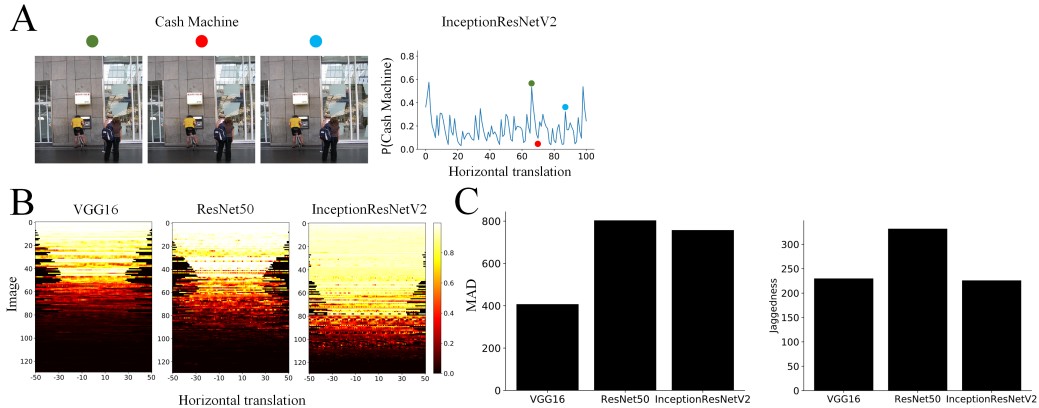

Figure 8: Modern deep convolutional neural networks are sensitive to small image translations (Without image downscaling). A) Example of InceptionResNetV2 sensitivity to very small horizontal translations. B) Comparison of three networks of various depths (16, 50, and 134 for VGG16, ResNet50, and InceptionResNetV2 respectively) in the task of horizontal image translation. Rows are sorted by sum. B) Modern networks have more jagged predictions. Jaggedness is calculated by counting the number of times the network's predictions had the correct class in its top-5 and after just one pixel shift it moved outside of the top-5 (and also the opposite transition from non-top-5 to top5). Similar results can be seen with the alternative measure of jaggedness: Mean Absolute Difference (MAD)

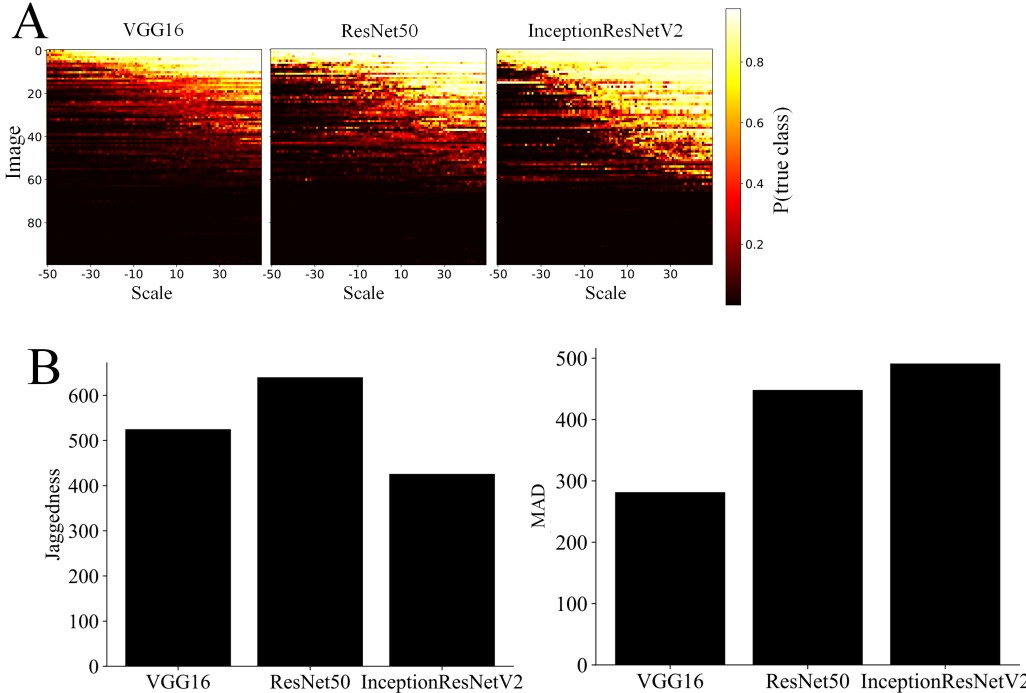

Figure 9: Modern deep convolutional neural networks are sensitive to small image rescalings A) Comparison of three networks of various depths (16, 50, and 134 for VGG16, ResNet50, and InceptionResNetV2 respectively) in the task of image rescaling. Rows are sorted by their sum. B) Quantification of the jagged behaviour of the deep neural networks.

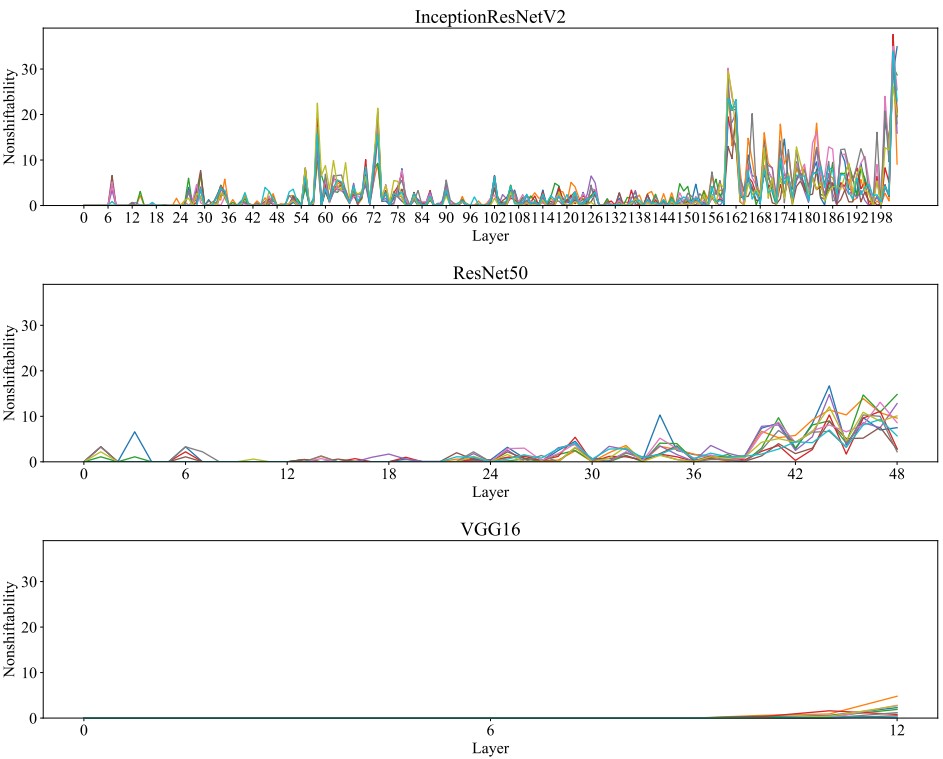

Figure 10: Nonshiftability as a function of depth in the three networks. Nonshiftability is defined as the number of times the global sum of a feature map changes by more than 20% of the mean as the input is translated. We only consider feature maps where the maximum response was above a threshold. According to our analysis, this measure should be zero if the representation is shiftable. Each line shows the nonshiftability in different layers in response to a randomly selected image.

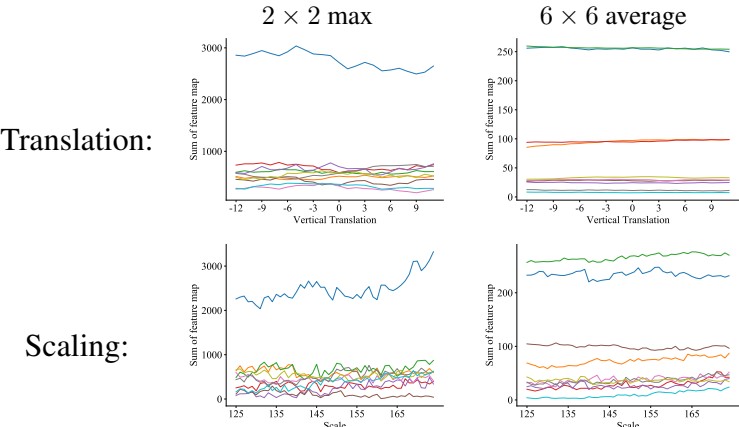

Figure 11: Another randomly selected image. Plotted are summation of feature maps from the last layer of VGG (spatial dimension 7x7) as the input is vertically translated in the image plane (top), or rescaled (bottom). Left: the original VGG16 with its 2x2 max pooling layers. Right: VGG16 where every 2x2 max pooling layer was replaced by a 6x6 average pooling layer.

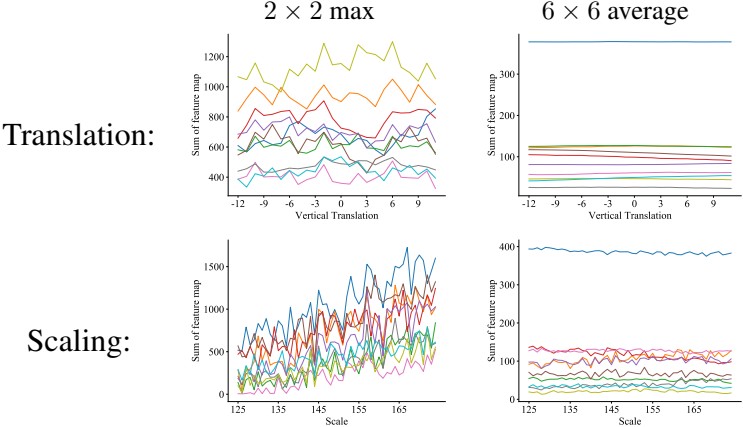

Figure 12: Another randomly selected image. Plotted are summation of feature maps from the last layer of VGG (spatial dimension 7x7) as the input is vertically translated in the image plane (top), or rescaled (bottom). Left: the original VGG16 with its 2x2 max pooling layers. Right: VGG16 where every 2x2 max pooling layer was replaced by a 6x6 average pooling layer.

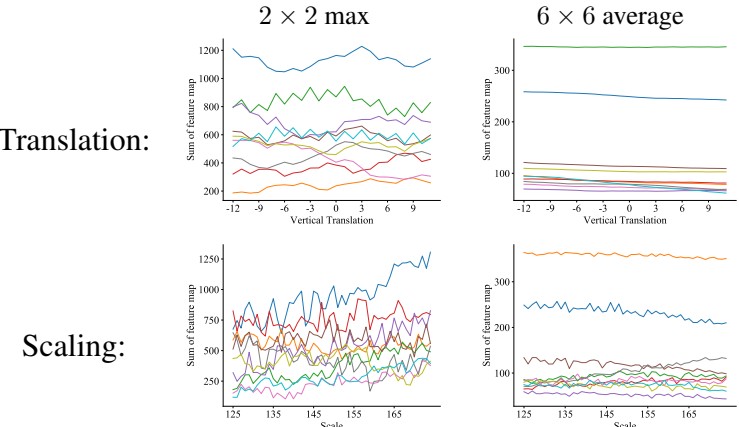

Figure 13: Another randomly selected image. Plotted are summation of feature maps from the last layer of VGG (spatial dimension 7x7) as the input is vertically translated in the image plane (top), or rescaled (bottom). Left: the original VGG16 with its 2x2 max pooling layers. Right: VGG16 where every 2x2 max pooling layer was replaced by a 6x6 average pooling layer.

