# OpenReview forum: "Why do deep convolutional networks generalize so poorly to small image transformations?"
_ICLR.cc/2019/Conference_

### Official Review · AnonReviewer2 · 2018-10-29
**Although the general idea is interesting, experimental evaluation is not convincing. Similarly, some explanations like the photographer's bias as reason for susceptability to very small image transformation is not entirely convincing.**

**Rating:** 5
**Confidence:** 4

**Review:**

Paper summary:

As is made clear in the title, this paper sets out to answer the following question: “Why do deep convolutional networks generalize so poorly to small image transformations?”. It focuses on natural image transformations on translation and scaling (rotation is missing though).

The paper proposes two main explanations:
-	Strided convolution, called subsampling in the paper, ignores the classical sampling theorem,
-	CNNs will not learn invariance because of the (photographers') biases contained in the datasets.

On a general level, the paper is a good read, it is well written and the figures clearly convey the message they’re intended to. Adversarial attacks and robustness of CNNs in general is a very interesting and important topic in ML. The originality of this work is in the approach of the problem, the paper tries to explain the reasons why CNNs are vulnerable. Related works put more emphasis on coming up with novel attacks/defense strategies. Considering natural attacks as done in this submission is particularly interesting as it is probably a more surprising shortcoming of CNNs compared to optimally designed attacks or highly unnatural perturbations. The argument about subsampling (stride) being the reason of not having translational invariance is nice, especially the theoretical insight with the Shannon-Nyquist theorem and the more figurative example on part detectors. There are nevertheless a few major concerns about this work:

Major Concerns:

Theoretical arguments:
The theoretical argument made in this paper is interesting but to make the point stronger a more in-depth explanation would be needed.
-	The step from Eq (2) to Eq (3) is not entirely clear “K does not depend on x_i”, maybe one extra sentence to explain this step would be useful.
-	Terms introduced such as the basis function B and the set of transformations T could be better defined.
-	For the extension to other types of transformations “While the claim focuses on global translation, it can also be extended to piecewise constant transformations.” it would be important to point out what type of natural transformations can be included in this set.

Empirical evidence:
Experiments are not fully convincing. Additional empirical evidence would be beneficial and necessary to support the claims of this:
-	“A natural criticism of these results is that they are somehow related to the image resizing and inpainting procedures that we used.” This is a very good point and the authors following arguments are not fully convincing. Results with different transformation procedures mentioned in the rest of the paragraph (and probably more) should be included to convince the reader.
-	The theoretical argument that translation invariance is not guaranteed because of the stride (subsampling) is not fully convincing and needs further explanation and experimental verification. In fact, feature maps of the CNNs that the authors consider do indeed contain many high frequencies.
-	The argument made in part 4 about the photographer’s bias seems valid for general natural transformations, but it does not apply to small transformations such as 1-pixel translations presented in the paper. Also, evidence that datasets without (or less) photographers' bias are less susceptible to natural attacks would make the argument in the paper a lot stronger.
-	When using 6x6 avg pooling for the VGG16 architecture ”recognition performance decreases somewhat” . Results are only preliminary in the paper, but this statement needs a more thorough experimental backing. It should come with convincing quantitative evidence.
-	Please include some results or citation on other work about test time augmentation to support the statement “still only provides partial invariance”.

References and phrasing:
Generally previous work is well referenced in this paper, although there are some formulations that can be slightly modified to make a clear distinction between what is novel and what is previous work:
-	As is very well shown in the introduction, there is a lot of work on generating adversarial examples that drastically change the output of a CNN. This should be made clear in the abstract, in fact the sentence “In this paper we show that modern CNNs [...] also happens with other realistic small image transformations”  seems to indicate that this is the novel work in the paper. This is also why I believe the first sentence “Deep convolutional network architectures are often assumed to guarantee generalization for small image translations and deformations.” is somewhat contestable.
-	“We find that modern deep CNNs are not invariant to translations, scalings and other realistic image transformations” as the paper points out earlier this is not a novel finding, so I would use a formulation that makes that clear and gives more emphasis to your own arguments as of why this happens.

Further Comments :
-	Part 5 "Implications for Practical Systems" could be moved to discussion as there is no new point and it seems more a reflection on what was already stated.
-	The final sentence of the abstract “Taken together our results suggest that the performance of CNNs in object recognition falls far short of the generalization capabilities of humans.” is not necessary, this is clearly true but it isn’t really contested in the ML community.
-	“despite the architecture being explicitly designed to provide such invariances” I agree that this has motivated the use and design of CNNs in the first place, but modern architectures are mostly designed to surpass the results on the common benchmarks rather than to provide such invariances.
-	”jaggedness is greater for the modern, deeper, networks compared to the less modern VGG16 network” might be worth interesting to consider if the residuals have anything to do with it.

---

> ### Author Response · Authors · 2018-11-21
> **Response**
>
> Thank you for the helpful comments.
>
> "Theoretical arguments". We will clarify the proof and notation in the next version. Regarding piecewise constant transformations: many interesting transformations (e.g. small rotations and scalings) can be approximated by a piecewise constant transformation. Figure 4 in the paper shows that scalings of the image behave as one would expect according to this extension of our analysis.
>
> “The theoretical argument that translation invariance is not guaranteed because of the stride (subsampling) is not fully convincing and needs further explanation and experimental verification”
>
> We prove in the paper (observation at bottom of page 4) that networks with no stride will be translation invariant. The proof ignores edge effects but we have also verified empirically that networks with the same architectures that are used in the modern CNNs but with no stride are invariant to the range of  translations that we consider in the paper. We will mention this empirical verification in the next version.
>
> "Empirical evidence:". As mentioned in our comments to R1, we now have quantitative results (jaggedness and recognition performance) for VGG16 and ResNet50 trained with larger pooling regions and we will include this in the final version. Also, as mentioned in our response to R1, we tried three different types of transformations: inpainting (figure 2), cropping (figure 8) and black border (not shown). All three gave the same results.
> Regarding the photographer's bias, we believe the presence of this strong bias allows the network to do well in the training set and test set without learning to be invariant to all single pixel shifts (without the bias, the number of shifts to memorize would be prohibitive). We will clarify this in the final version.
> We very much agree that "feature maps of the CNNs that the authors consider do indeed contain many high frequencies.” and our paper argues that this aliasing is the reason for the lack of invariance: we show analytically that if the feature maps did not contain high frequencies then global pooling would be invariant, and we show empirically that feature maps of modern CNNs do not obey the conditions required for shiftability (figure 10).
>
>
> "References and phrasing:". Thank you again for these comments. We will revise the final version accordingly.

---

### Official Review · AnonReviewer1 · 2018-11-01
**Nice empirical study of invariants in modern CNNs, with quantitative support at all key points.**

**Rating:** 7
**Confidence:** 4

**Review:**

This paper describes an empirical study of translation and scale
invariance properties of modern CNN architectures. The authors conduct
a thorough study of translation invariance in VGG16, ResNet50, and
InceptionResNet with respect to the Nyquist frequency and shift-
versus translation-invariance properties of network layers as a
function of depth and subsampling rate. Empirical observations are
quantified using a variety of metric to measure the stability of
feature maps under geometric transformations of the input.

The paper has the following strong points:

 1. It tells an interesting (and engaging) story about a largely
    empirical study, and while doing this never pretends to be more
    than it is.
 2. Empirical observations are supported by quantitative measures that
    give compelling evidence for most observations in the paper. The
    discussion about shiftability versus translatability is
    particularly interesting with its link to nonlinearity, smoothing
    and Nyquist limits.

The paper has the following weak points:

 1. The reliance on inpainting for almost all experiments is somewhat
    worrying. It is not clear that this procedure isn't introducing
    its own biases affecting translation and scale invariance. The
    authors make reference to a separate protocol reported in the
    appendices, but it isn't clear which results in the appendices
    they are referring to. A more thorough control study seems in
    order to verify that inpainting is a reasonable simulation.
 2. Some figures are scaled down to the limits of legibility.

In summary: I like this paper a lot, and I think it adds useful
elements and analytical tools (both theoretical and empirical) to the
discussion on invariants in modeern CNNs.

---

> ### Author Response · Authors · 2018-11-21
> **Response**
>
> Thank you for the helpful comments.
>
> To establish that our results are not a side effect of our inpainting setup we repeated the same experiment with three different setups: cropping the image (figure 8 from the appendix), inpainted background (figure 2), and black background (not shown). All setups give qualitatively similar results, suggesting that the phenomenon we observe isn’t a mere artifact of the choice of setup.
> We thank the reviewer for pointing out that are figures are too small and we will fix this on the next version.

---

### Official Review · AnonReviewer3 · 2018-11-04
**Critical topic, but limited novelty and results**

**Rating:** 7
**Confidence:** 5

**Review:**

This paper studies the lack of shift invariance in state-of-the-art neural networks, namely, the paper introduces results that show that state-of-the-art deep neural networks are affected by 1-pixel shifts because the convolutional layers in the network poorly sample the feature maps. The topic addressed in the paper is critical for many computer vision systems, as lack of shift invariance is a catastrophic failure mode. The arguments of the paper help clarifying why the networks are so sensitive to small shifts of the objects (poor subsampling) but generalize well (there is a bias in the location of the objects in the dataset).  Both of these arguments and the sensitivity of the networks to small shifts are well known in the literature, but it is great to see a paper that puts them together and tests these arguments in state-of-the-art deep nets for ImageNet.

However, the paper could do a much better job providing evidence to support the arguments:

*Few quantiative results on the sensitivity to 1-pixel shift, most results are qualitative. This makes hard to assess whether the reported results are "accidents" found in certain images or are general. The results that support that 1-pixel shifts affect state-of-the-art neural networks are in Figure 2b. Yet, these results are unclear, eg. is "400 Jaggedness" a lot?, what is the size of the embedded image?, How are the 100 images selected? Is the network performing well in those images? How does the size of the embedded image change the "Jaggedness"?

*Something that could help to strengthen the results would be to add networks with better sampling + larger pooling regions and see how this solves the lack of shift invariance. Now it has only been tested increasing the pooling regions, which misses the main point of the paper. Also, the results on these networks with larger pooling regions, are all qualitative.

*The mathematical proof is done for average pooling, which is rarely used nowadays. I would suggest using max pooling. Also, the aforementioned experiment in which the size of the pooling region is increased, is it max pooling or average pooling?

*Limited results on the ImageNet bias. These results are reported in one image category (Figure 5), how general are them?

*The paper assumes that shifting an image embedded an object in a black background is equivalent to shifting an object in a static background. A hypothesis would be that the embedding of the image in the black background creates artificial boundaries that make the network more fragile to 1-pixel shifts than for natural images.

In summary, I think this is a paper that may arise a lot of interest, although the different arguments are known and the experiments are poorly executed.

---

> ### Author Response · Authors · 2018-11-21
> **Response**
>
> Thank you for the useful comments.
>
> “hard to assess whether the reported results are "accidents" found in certain images or are general. “
>
> As we write in the text (third paragraph, page 3)  for approximately 30% of the randomly selected images, there exists a one pixel shift that removes the correct prediction from the top-5 (or vice versa).  Alternatively, we can measure the number of one pixel shifts that change the top 1 prediction of the network, and here the numbers are even larger: for a randomly selected image from ImageNet  embedded at a random vertical location and a fixed horizontal location,  a one pixel vertical translation will change the prediction of the InceptionResNetV2 network with a probability greater than 25%. Yet another way of saying the same thing: For a randomly selected image from ImageNet, we found on average 18.5 (25% of 74 translations) different locations where we can vertically translate the location of the embedded image by one pixel and get a change in prediction.
>
> These are clearly general results, not accidents. We will add the top-1 numbers to the next version.
>
> "Few quantiative results on the sensitivity to 1-pixel shift, most results are qualitative."
>
> Results in figure 2 show the results on 100 randomly selected images and for each one we tested 74 different 1-pixel shifts so figure 2 summarizes 7400 different experiments for each architecture. Similarly, figures 8 and 9 summarize another 7400 different experiments for each architecture.
>
> As we write in the text (caption of figure 2) "Images (rows) are randomly chosen from the ImageNet dataset" and indeed we verified that performance on this subset of images is about the same as the overall reported performance of each architecture. The size of the embedded image is 150 and figure 6 shows how the size of the embedded image affects performance. Following the reviewer’s suggestion, we measured the effect of the embedding size on the jaggedness and found that smaller size increases jaggedness as expected. We will add this to the next version of the paper.
>
>
> "results on these networks with larger pooling regions, are all qualitative."
>
> We now have quantitative results (jaggedness and recognition performance) for VGG16 and ResNet50 trained with larger pooling regions (accuracy goes down by about a factor of two, as does jaggedness). We will include these in the final version.
>
> "The mathematical proof is done for average pooling, which is rarely used nowadays".
>
> To our knowledge, *global* average pooling is the common choice in modern CNNs. One example from the ResNet paper: “The network ends with a global average pooling layer and a 1000-way fully-connected layer with softmax.”
>
> “Also, the aforementioned experiment in which the size of the pooling region is increased, is it max pooling or average pooling?”
> Average pooling. From the captions of figure 4: "Right: VGG16 where every 2x2 max pooling layer was replaced by a 6x6 average pooling layer."
>
> "Limited results on the ImageNet bias. These results are reported in one image category (Figure 5), how general are them? "
>
> As written in the paper, we have actually performed this experiment on all ImageNet categories. From page 7: "To be more quantitative, we used the available bounding-box labels, and extracted the center point of the bounding-box and its height as proxies for the object position and size respectively. We then applied a statistical significance test to ask whether object location and object sizes were uniform for that category. For more than 900 out of the 1000 categories we found that location and size were highly non uniform"
>
> "The paper assumes that shifting an image embedded an object in a black background is equivalent to shifting an object in a static background."
>
> Note that we do not use a black background, but rather we use an inpainting procedure to precisely address the reviewer's concern and make sure that no artificial boundaries are created. We also show (figure 8) that results are similar when we use cropping from a larger image, i.e. without inpainting or black borders. We have also run the experiments with black background and achieved similar results.

---

> > ### Comment · AnonReviewer3 · 2018-11-30
> > **Thanks for the clarifications.**
> >
> > Thanks for the clarifications.

---

### Public Comment · (anonymous) · 2018-11-16
**Problems with reproducing the result**

We attempted to reproduce the jagged predictions seen in this work's Figure 1, using the polar bear video. The paper claims that the video, when naturally classified, rapidly changes in prediction confidence and often misclassifies frames, i.e., labels them as other than "polar bear."

We were unable to reconstruct these results. We found that the classification pipeline -- with the same classifier and video as in the paper -- correctly classifies the frames as polar bear in a smooth and reliable fashion.

The resulting video can be found here: https://youtu.be/NF4hc-RiXqg

Based on examining the frames presented in the paper and the subsequent correspondence with the authors, we are led to believe that the authors missed a crucial part of the image classification pipeline: square cropping (by the shortest edge length) the image before resizing. In fact, in Figure 1, we can see how the polar bear looks elongated vertically.

------------------------
Technical details:

In our reproduction we used the exact same video and network that the paper claims to use, the InceptionResNet-V2 model from Keras. To understand why our classification pipeline worked and this paper's authors' pipeline did not, we step back and examine the pipeline for image classification. For some image x we:

1. Take image x, then center square crop based on the shortest side of the image
2. Resize this square, cropped image to 299x299 pixels
3. Perform preprocessing on input
4. Feed preprocessed inputs to model for classification

We performed all these steps correctly and thereby achieved smooth, correct predictions. Based on the correspondence with the authors, we are led to believe that they did not perform the crucial cropping step in part 1 - they only scale the original image to 299x299 pixels, then proceed to step 3.

---

> ### Author Response · Authors · 2018-11-16
> **These problems have already been addressed in the text**
>
> Thank you for your comment. Indeed we had a correspondence about this issue following a posting of a previous version of our paper on ArXiv.
>
> The reviewer is referring to one particular result shown at the bottom of figure 1. Contrary to the title of the reviewer’s comment, he/she acknowledged in our correspondence that given the input frames shown in the bottom of figure 1, the output of the network that we show is reproducible and it varies drastically even between almost identical frames.
>
> The issue that the reviewer is addressing is how to go from the YouTube video to the network input since the video has an aspect ratio very different from one. Indeed for this specific video, this “processing pipeline” makes a difference. However, and as we already mentioned in the correspondence, we easily found other YouTube videos of polar bears that give jagged predictions with the reviewer’s suggested pipeline.
>
> As a result of the same correspondence, we added a footnote at the bottom of page 3 that explicitly says that different preprocessing pipelines give different results.
> Importantly, following the reviewer's comments, we verified that all the other results do not depend on the pipeline and all the graphs shown in the paper are with a pipeline that maintains the aspect ratio of the original image.
>
> That being said, the question of "the right pipeline" is far from obvious. Using a center crop as suggested by the reviewer means that you will necessarily miss polar bears that do not appear in the center (an extreme case of taking advantage of photographer's bias). In fact, the default preprocessing suggested by Keras (https://keras.io/applications/)  prefers to reshape the full image (as we did in the bottom of figure 1) rather than to perform a center crop.
>
> Finally, we want to stress again that the three frames shown in the bottom of figure 1 are exactly the frames given to the network and given these frames anyone can reproduce our reported network output. The difference between these frames is a very small deformation that is imperceptible to humans and yet makes a huge difference to the output of the network. The goal of our paper is to explain why this happens.

---

> > ### Public Comment · (anonymous) · 2018-11-19
> > **Response**
> >
> > Thank you for your response.
> >
> > The input processing pipeline of Inception-ResNet-v2 (and indeed every Inception family classifier), as per the code release of the original paper, indicates that original Inception-ResNet-v2 model was trained such that it would be evaluated with a pipeline including a center crop (see *** below). Therefore since there is a shift in the distribution of input images due to preprocessing differences we cannot conclude that the observed 'jaggedness' is inherent to the model (in the authors results that use this pipeline (including Figure 1)).
> >
> > We are not debating the fact that the input pipeline change can lead to incorrect predictions. We are also not debating that the frames in the figure are the frames given to the network. All we are saying is that these frames are distorted in a way that the network is not trained to expect. You can see a side by side comparison here: https://i.imgur.com/TxJeCsx.png.
> >
> > Therefore, it is unclear how the results presented in the paper support the claim that the standard classifier fails on natural images. It is hard to expect the classifier to work  when the input has (in effect) been transformed in an unexpected way. (After all, once the deformation introduced by misconfiguration of the pipeline are corrected, the model classifies the resulting images correctly.)
> >
> > Regarding correspondence, we were not shown any videos that induced jagged predictions when classified with the correct input processing pipeline (although these may exist, and we would be happy to evaluate any source videos that the authors give -- we were unable to find the source videos for the other videos shown in Figure 1).
> >
> > ------------------------
> > ***
> > During Inception-ResNet-v2 evaluation, as per the paper and its authors code release (https://github.com/tensorflow/models/tree/696b69a498b43f8e6a1ecb24bb82f7b9db87c570/research/slim), we can see that inference preprocessing uses a center crop. Any other input pipeline configuration is incorrect.
> >
> > Evaluation calls the function `preprocess_image` here:
> > https://github.com/tensorflow/models/blob/master/research/slim/preprocessing/inception_preprocessing.py#L285
> >
> > Which ultimately uses a center crop, as seen here in the `preprocess_for_eval` function:
> > https://github.com/tensorflow/models/blob/master/research/slim/preprocessing/inception_preprocessing.py#L244

---

> > > ### Author Response · Authors · 2018-11-20
> > > **misunderstanding?**
> > >
> > > Thank you for your comment.
> > >
> > >
> > > Perhaps it makes sense to take this particular discussion about videos offline since we do not seem to be converging despite many rounds of back and forth (maybe a phone call will be helpful).
> > >
> > > The important thing to remember is that except for the bottom of Figure 1, all the other results use ImageNet images,  NOT YouTube videos. Thus the statement in the reviewer's comment "we were unable to find the source videos for the other videos shown in Figure 1" seems to suggest a misunderstanding by the reviewer. As the text clearly describes, we took an image from imageNet, resized it (while preserving aspect ratio) , embedded it in a square image that is the size expected by the network and synthetically moved it around or rescaled it. As we also explain in the text, we experimented with different ways to embed the image into the square image (black background, inpainting, cropping) and we found a lack of invariance using all these hundreds of images
> > >
> > > The following anonymous dropbox folder contains YouTube videos that use exactly the same pipeline as suggested by the reviewer and the network outputs are highly jagged.
> > > https://www.dropbox.com/sh/wvis40zd3sltf31/AACn480D7DB14FtneBaIMWPoa?dl=0
> > >  We are going to upload a code to github with the original frames to reproduce these results. We are also happy to send it to the reviewer.
> > >
> > >  Again, we emphasize that figures 2 and 8 in the paper summarize experiments with thousands of presentations of images to the different networks, and all of these images are from ImageNet and do not require the reviewer's suggested pipeline nor the pipeline we describe in the appendix.

---

### Author Response · Authors · 2019-12-25
**A new version of the paper**

A new version can be found in the following link:

http://jmlr.org/papers/v20/19-519.html

---

### Meta-Review · Area_Chair1 · 2018-12-14
**empirical study of invariance on modern CNNs**

**Confidence:** 5
**Recommendation:** Reject

**Metareview:**

This paper attempts to answer its suggestive title by arguing that this generic lack of invariance in large CNN architectures is due to aliasing introduced during the downsampling stages.
This paper received mixed reviews. Positive aspects include the clarity and exhaustive empirical setups, whereas negative aspects focused on the lack of substance behind some of the claims. Ultimately, the AC took these considerations into account and made his/her own assessment, summarized here.

The main claim of this paper implies the following: modern CNNs are unable to build invariance to small shifts, but somehow are able to learn far more complex invariances involving lighting, pose, texture, etc. This must be empirically verified beyond reasonable doubt, and the AC thinks that the current experimental setup does not achieve this threshold. As mentioned by reviewers and by public comments, the preprocessing pipeline is a key factor that may be confounding the analysis, and this should be better analysed. For example, as mentioned in the reviews below, the shift in the image can be either done by inpainting, cropping, or using a fixed background. The authors claim that there are no qualitative differences between those preprocessing choices, but by inspecting Figures 2B and 8C, the AC notices a severe change in 'jaggedness'; in other words, the choice of preprocessing *does* affect the quantitative measures of (un)stability, even though the qualitative assessment (unstable in all setups) is the same. In particular, using non-centered crops should be the default setup, since it requires no preprocessing. It is confusing that it appears in the appendix instead of the inpainting version of figure 2b. This is important, since it implies that the analysis is mixing two perturbations: the actual action of the translation group and the choice of preprocessing, and that the latter is by no means negligible. I would suggest the authors to perform the following experiment to disentangle the effect of translation by the effect of preprocessing. Since the translation forms a group, for any shift applied to the image, one can 'undo' it by applying the inverse shift. Say one applies a shift to image x of d pixels and obtains x'=T(x,+d) as a result (by using whatever border handling procedure). If border effects were negligible, then x''=T(x',-d) should give us back x, so a good measure of how unstable the network is is to measure the difference in prediction between x,x' and x''. If predicting x'' is as unstable as predicting x', it follows that the network is actually unstable to the border effect introduced by T.

Given this, the AC recommends rejection at this time, and encourages the authors to resubmit their work by addressing the above point.